# The sugar content of foods in the UK by category and company: A repeated cross-sectional study, 2015-2018

**Lauren K. Bandy**[1]*, **Peter Scarborough**[1], **Richard A. Harrington**[1], **Mike Rayner**[1], **Susan A. Jebb**[2]

**1** Nuffield Department of Population Health, University of Oxford, Oxford, United Kingdom, **2** Nuffield Department of Primary Care Health Sciences, University of Oxford, Oxford, United Kingdom

* lauren.bandy@ndph.ox.ac.uk

**Data Availability Statement:** This study used data from two commercial sources. The sales data was accessed under licence from Euromonitor International (https://www.euromonitor.com/

## Abstract

### Background

Consumption of free sugars in the UK greatly exceeds dietary recommendations. Public Health England (PHE) has set voluntary targets for industry to reduce the sales-weighted mean sugar content of key food categories contributing to sugar intake by 5% by 2018 and 20% by 2020. The aim of this study was to assess changes in the sales-weighted mean sugar content and total volume sales of sugar in selected food categories among UK companies between 2015 and 2018.

### Methods and findings

We used sales data from Euromonitor, which estimates total annual retail sales of packaged foods, for 5 categories—biscuits and cereal bars, breakfast cereals, chocolate confectionery, sugar confectionery, and yoghurts—for 4 consecutive years (2015–2018). This analysis includes 353 brands (groups of products with the same name) sold by 99 different companies. These data were linked with nutrient composition data collected online from supermarket websites over 2015–2018 by Edge by Ascential. The main outcome measures were sales volume, sales-weighted mean sugar content, and total volume of sugar sold by category and company. Our results show that between 2015 and 2018 the sales-weighted mean sugar content of all included foods fell by 5.2% (95% CI −9.4%, −1.4%), from 28.7 g/100 g (95% CI 27.2, 30.4) to 27.2 g/100 g (95% CI 25.8, 28.4). The greatest change seen was in yoghurts (−17.0% [95% CI −26.8%, −7.1%]) and breakfast cereals (−13.3% [95% CI −19.2%, −7.4%]), with only small reductions in sugar confectionery (−2.4% [95% CI −4.2%, −0.6%]) and chocolate confectionery (−1.0% [95% CI −3.1, 1.2]). Our results show that total volume of sugars sold per capita fell from 21.4 g/d (95% CI 20.3, 22.7) to 19.7 g/d (95% CI 18.8, 20.7), a reduction of 7.5% (95% CI −13.1%, −2.8%). Of the 50 companies representing the top 10 companies in each category, 24 met the 5% reduction target set by PHE for 2018. The key limitations of this study are that it does not encompass the whole food market and is limited by its use of brand-level sales data, rather than individual product sales data.

packaged-food) via the Bodleian Library, University of Oxford, using Euromonitor's database portal Passport GMID. The product information dataset, including nutrition composition data, was purchased for the purpose of the lead author's DPhil research project from Edge by Ascential (https://www.ascentialedge.com/our-solutions). Due to licencing restrictions, the Euromonitor and Edge by Ascential datasets can only be requested under licence for the purpose of verification and replication of study's findings via the research group's Data Access Committee (contact: Trisha Gordon foodDBaccess@ndph.ox.ac.uk). Further use of these datasets must be negotiated with the data owners (Euromonitor contact: Ashton Moses - passport.support@euromonitor.com, Edge by Ascential contact: David Beech - info@ascentialedge.com).

**Funding:** LB and MR are funded by the Nuffield Department of Population Health, University of Oxford. PS is funded by a British Heart Foundation Intermediate Basic Science Research Fellowship (FS/15/34/31656). All authors are part of the National Institute for Health Research (NIHR) Oxford Biomedical Research Centre (BRC). SJ is also funded by the NIHR Collaboration for Leadership in Applied Health Research and Care Oxford at Oxford Health NHS Foundation Trust and is an NIHR senior investigator. The funders had no role in study design, data collection and analysis, decision to publish, or preparation of the manuscript.

**Competing interests:** The authors have declared that no competing interests exist.

**Abbreviations:** PHE, Public Health England.

## Conclusions

Our findings show there has been a small reduction in total volume sales of sugar in the included categories, primarily due to reductions in the sugar content of yoghurts and breakfast cereals. Additional policy measures may be needed to accelerate progress in categories such as sugar confectionery and chocolate confectionery if the 2020 PHE voluntary sugar reduction targets are to be met.

## Author summary

### Why was this study done?

- Sugar intakes around the world exceed dietary recommendations, and this increases the risk of excess energy intake and weight gain, diabetes, and dental caries.

- In an attempt to reduce sugar consumption, the UK government has set voluntary 5% and 20% sugar reduction targets for industry to achieve by 2018 and 2020, respectively.

- This study was conducted to see how the sales-weighted mean sugar content of individual companies' product profiles changed between 2015 and 2018.

### What did the researchers do and find?

- Researchers analysed the sales-weighted mean sugar content of products in the 5 food categories that contribute the most to sugar intake in the UK: biscuits and cereal bars, breakfast cereals, chocolate confectionery, sugar confectionery, and yoghurts.

- Overall, the sales-weighted sugar content of these products fell by 5%, from 28.7 g/100 g in 2015 to 27.2 g/100 g in 2018, with the largest reductions seen in yoghurts (−17%) and breakfast cereals (−13%).

- Of the 50 companies representing the top 10 companies in each category, 24 met the 5% sugar reduction targets for 2018.

### What do these findings mean?

- Our findings show that there has been a small reduction in the sugar content of foods in the UK, and approximately half of companies had not met the 5% sugar reduction target by 2018.

- Additional policy measures may be needed to further reduce the sugar content of these foods.

## Introduction

One in 5 deaths globally are linked to a poor diet [1]. High consumption of free sugars is associated with increased energy intake and weight gain [2], type 2 diabetes [3], and dental caries [4]. In 2015, the World Health Organization called on countries to reduce the sugar intakes of both adults and children, recommending that the intake of free sugars not exceed 5% [5]. This was followed by similar advice from the UK Scientific Advisory Committee on Nutrition, which recommended a target for dietary energy intake from free sugars of 5% [6]. According to national dietary surveys, the consumption of free sugars in the UK is twice the guideline intake for adults and almost triple for children aged 4–18 years [7]. Citing the success of the salt reduction targets, and in a further effort to reduce the population's consumption of sugars, in March 2017 Public Health England (PHE), an executive agency of the UK Department of Health and Social Care, outlined a series of voluntary sugar reduction targets for businesses. A 20% sugar reduction target was set for 9 food categories by 2020, with an interim 5% target for 2018, based on the sales-weighted mean sugar content of products in 2015. Sugar reduction may be achieved by a variety of methods, including reformulating existing products, shifting sales between high- and low-sugar products, and launching new products into the marketplace [7]. The 9 categories covered by PHE's targets represent 54% of sugar consumed by children aged 4–10 years, and 34% for adults aged 19–65 years [7]. This initiative is part of a wider sugar reduction programme, including the introduction of the Soft Drinks Industry Levy in 2018 [8], public health awareness campaigns such as Change4Life [9], and increased attention to free sugar and its health impacts in the mainstream media [10].

In a time when some countries, states, and cities are implementing mandatory nutrition policies, including taxes on soft drinks [11] and front-of-package warning labels [12,13], there is much interest and debate around whether voluntary initiatives are effective. The success of PHE's voluntary sugar reduction policy is heavily dependent on action by the entire food industry to reduce the sugar content of its products and to encourage changes in consumer behaviour towards purchasing lower sugar alternatives by launching new product lines or focussing marketing and advertising practices on lower sugar products. It is therefore important to analyse the actions of individual companies in relation to sugar reduction. PHE has previously published an evaluation of this policy that analysed category-level changes in sales-weighted mean sugar content over the same time period (2015–2018) [14]. It found that the sugar content of products fell by −2.9%, with total volume sales of sugar increasing by 2.6%. The PHE report included only a limited number of companies, there was no indication of the variability between companies, and the data were not peer-reviewed.

The aim of this study was to use alternative and more comprehensive datasets with information on the nutrient composition of foods and food sales data to analyse how the sales-weighted mean sugar content and total volume of sugars sold from foods covered by PHE's sugar reduction targets have changed by category and company between 2015 and 2018.

## Methods

This study was not conducted as part of any preplanned analyses. It was undertaken as part of a DPhil (PhD) project, with analyses being carried out between March and October 2019. All sensitivity analyses were added during the peer-review process. This study is reported as per the Strengthening the Reporting of Observational Studies in Epidemiology (STROBE) guideline (S1 STROBE Checklist).

## Data types and sources

Data on the sugar content of foods were sourced from a commercial third party, Edge by Ascential (previously known as Brand View). Edge by Ascential collects product information, including nutrient composition data, by scraping the websites of 3 leading UK retailers: Tesco, Sainsbury's, and Asda. The product information used in this study was collected on the same date (13 December) in 4 consecutive years (2015–2018), with 2015 being as far back historically as data were available.

Data were provided for all food and beverage products. Each product contained the following: date and year, retailer name, product name, brand name, company (manufacturer) name, barcode, price, ingredients, pack size, serving size (if stated), and nutrient composition per 100 g for energy, protein, carbohydrate, sugars, fat, saturated fats, fibre, and salt. Each product was also assigned to a product category. These categories were used to identify the relevant products to be included in this study. Only 5 categories from PHE's sugar reduction targets (biscuits and cereal bars, breakfast cereals, chocolate confectionery, sugar confectionery, and yoghurts) were easily identifiable. Duplicates sold in different supermarkets were removed based on barcode and year.

Weighting composition data by sales volume indicates what was sold, which is a better proxy for consumption and more relevant to public health than analysing the nutrient content of available products only. Therefore, each individual product in the Edge by Ascential database was matched with sales data sourced from Euromonitor via the Bodleian Library, University of Oxford. Euromonitor is a private market research company that provides sales data collected from primary and secondary data sources, including store audits, interviews with companies, publicly available statistics, and company reports [15]. The Euromonitor dataset used in this study includes brands sold through all retail channels, including supermarkets, discount stores, convenience stores, traditional markets, and vending machines. This dataset did not include food service, and only approximately 85% of retail sales are covered, meaning the dataset does not represent the whole food market.

Five categories were identified where the sales database could be directly related to the PHE targets (Table 1). All of the sub-categories presented in Table 1 were included in this analysis. The sales database did not provide enough granularity to identify puddings, cakes, morning goods, and sweet spreads, which were therefore excluded.

Euromonitor measures sales by brand, rather than by individual products. A brand was defined as a set of products that have the same core name and are manufactured by the same company. For example, the company Mondelez manufacturers multiple brands, including Cadbury Dairy Milk chocolate (one brand) and Oreo biscuits (another brand). Within each

**Table 1. Number and list of categories and sub-categories included in analysis.**

| Public Health England food category | Number of Euromonitor sub-categories included in analysis | Euromonitor sub-categories included |
|---|---|---|
| Biscuits and cereal bars | 7 | Cereal bars, chocolate coated biscuits, cookies, filled biscuits, plain biscuits, snack bars, wafers |
| Breakfast cereals | 5 | Children's breakfast cereals, flakes, hot cereals, muesli and granola, other ready to eat cereals |
| Chocolate confectionery | 5 | Boxed assortments, chocolate pouches and bags, chocolate with toys, countlines (individual chocolate bars), tablets (large chocolate bars) |
| Sugar confectionery | 10 | Boiled sweets, liquorice, lollipops, mints, other sugar confectionery, pastilles, gums, jellies and chews, toffees, caramels and nougat |
| Yoghurts | 5 | Drinking yoghurt, flavoured fromage frais and quark, flavoured yoghurt, plain fromage frais and quark, plain yoghurt |

brand, there may be multiple individual products, for example Cadbury Dairy Milk Fruit and Nut and Cadbury Dairy Milk Whole Nut.

The brand-level sales data were matched with the product-level nutrient composition data based on the variables brand name, company name, category, and year. Where brands were matched with more than 1 individual product, a mean sugar content was calculated, as demonstrated in the flow chart in Fig 1 below.

Corresponding product-level nutrient composition data could not be found for 62 brands in the sales database for the 5 included categories. Twenty of these brands were not sold in the supermarkets included in the nutrient composition database. This represented 2% of total volume sales. For these cases, the nutrient composition data were sourced online from the brand website in mid-2019, and these data were used for all 4 years. Forty-two brands across the 5 categories, representing 3% of total volume sales, were human errors in the sales database and were not manufactured in the given time period. As these products could not be matched with corresponding nutrient composition data, they were removed from the dataset. Euromonitor classifies a number of small and local brands under the umbrella of 'others', and these products, representing 13.5% of total volume sales (ranging from 7% for sugar confectionery to 21.1% for biscuits and cereal bars), were also excluded from the main analyses; a sensitivity analysis was conducted to assess the impact on the results.

## Data analysis

The unit of analysis was the brand, and the sales-weighted mean sugar content (g/100 g) was the primary outcome, with total volume of sugars sold (tonnes) also being calculated, given

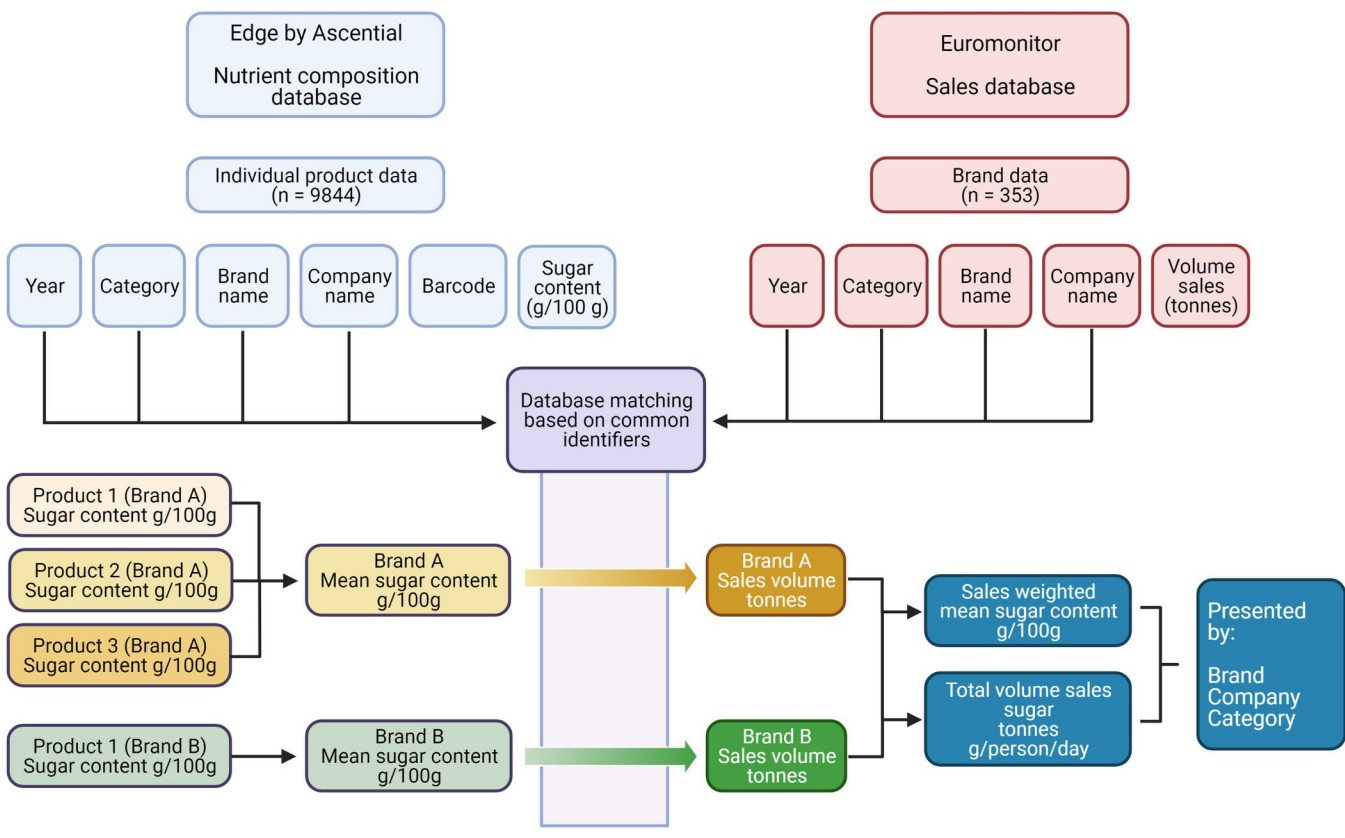

**Fig 1. Data and analysis flow chart.**

that it is a proxy for consumption and therefore adds important context for public health. The additional variables in the datasets identified the companies that owned the brands and the categories they belonged to, allowing for analyses to be stratified by company and category. Change in sales-weighted mean sugar content was calculated between 2015 and 2018, with all products being included regardless of whether or not they were present in the market in 2015 or 2018. To adjust for population changes and to put the results in the context of dietary recommendations, results for total volume of sugars sold are presented in grams per person per day. This value was calculated using annual population estimates from the Office for National Statistics [16], dividing by annual population size and by 365.

For each category, 5% and 20% sugar reduction targets were calculated using the sales-weighted mean sugar content with 2015 as a baseline.

The change in total volume of sugars sold was split into the change in the absolute mean sugar content and change in total volume sales using a decomposition formula [17]. Briefly, this is derived as follows. Total volume of sugar sold (V) = mean sugar content (S) × volume of foods sold (F). Differentiation (to give change in volume of sugar sold over time) gives $V' = F \times S' + S \times F'$. Let $\Delta V = V'/V$ (i.e., the annual percentage change in V). Then

$$\Delta V = \frac{F \times S' + S \times F'}{S \times F} = \frac{F \times S'}{F \times S} + \frac{S \times F'}{F \times S} = \frac{S'}{S} + \frac{F'}{F} = \Delta S + \Delta F$$

Therefore, the annual percentage change in the total volume of sugar sold is equal to the sum of the annual percentage change in the mean sugar content and the annual percentage change in the volume of the foods sold. This was calculated at the total, category, and company level, and percentage change was used to attribute absolute changes in the volume of sugar to changes in sugar content and changes in total volume sales.

For most categories, 95% confidence intervals were calculated using absolute and sales-weighted standard deviations and means using unique products identified in the nutrient composition database as the units of analysis. For sugar confectionery, the nutrient composition data were not normally distributed, so the mean and median sugar content and interquartile range were calculated. We tested for differences in the sales-weighted mean sugar content of each category in 2018 compared to 2015 using a Kruskal–Wallis test, weighted so that each brand was proportional to total sales.

## Sensitivity analysis

We conducted 2 sensitivity analyses. Sensitivity analysis 1 assessed the impact of individual product variation within brands on the overall sales-weighted mean sugar content of foods. To do this, brands within one category and year, e.g., breakfast cereals in 2018, were classified as either 'product range' brands, or 'individual product leader' brands. This was done based on the researcher's knowledge of the market and with the aid of product searches on online supermarkets. Product range brands are those that have a wide range of individual products and flavour variations but were judged to have no single product leader in terms of sales. These brands were excluded from the sensitivity analysis as the method of taking a mean sugar content of all individual products within a brand was unlikely to have affected the results. For brands that have a single individual product that is likely to represent the majority of that brand's sales (e.g., Original Weetabix) compared to other minority variants (e.g., Weetabix High Protein), assuming an equal weighting of sales amongst these individual products when calculating the mean sugar content is likely to have impacted the results. For brands classified as 'individual product leaders', we compared the sugar content of the leading individual

product with the sugar content values used for the main study (g/100 g) and calculated percentage difference and range.

Sensitivity analysis 2 was conducted to assess the impact of excluding the sales volume that represented small and local businesses under the umbrella term 'others'. Scenario 1 assumed that this volume had the same sales-weighted mean sugar content as the overall category in 2015 and remained unchanged over time. This scenario was considered most likely as small and local companies represented by the 'others' sales volume are less likely to have the capacity for reformulation. Scenario 2 assumed that the sales-weighted mean sugar content of the volume sales represented by 'others' changed at the same rate as the overall category between 2015 and 2018.

## Results

For 2015, 95 companies were included in this analysis (although, for clarity, only the top 10 companies by volume sales for each category are presented in Fig 3). The 95 companies manufactured 350 brands, with 2,515 products in the 5 included categories (biscuits and cereal bars, breakfast cereals, chocolate confectionery, sugar confectionery, and yoghurts) (Table 2). These figures remained relatively unchanged over the time period. In 2018, data were included for 97 companies producing 353 brands and 2,351 individual products.

The total volume of sales from the 5 food categories did not change between 2015 and 2018, with small increases in the volume sales of biscuits and confectionery counteracted by declines in the volume sales of breakfast cereals and yoghurts (Table 3).

The sales-weighted mean sugar content for included food categories fell from 28.7 g/100 g to 27.2/100 g, a reduction of 1.5 g/100 g, or 5.2% (95% CI −9.1%, −1.4%) although this change was not statistically significant ($p$ = 0.52) (Table 4). The greatest change was observed in yoghurts and breakfast cereals, with a reduction of 1.9 g/100 g, or 17.0% (95% CI −26.8%, −7.1%), and 2.5 g/100 g, or 13.3% (95% CI −19.2%, −7.4%), respectively. Biscuit and cereal bar mean sugar content declined by 2.5 g/100 g, or 6.3% (95% CI −10.0%, −2.7%), from 18.8 g to 16.3 g per 100 g. The reductions for chocolate (−1.0%) and sugar confectionery (−2.4%) were small.

The total volume of sugar sold decreased from 21.4 g/d to 19.8 g/d, a reduction of 1.6 g/d, or 7.5% (Table 5). Of this, 70% was attributable to a reduction in the mean sugar content of foods and 30% was due to a decrease in volume sales (Fig 2).

Reductions in the sugar content of products and the total volume of sugars sold varied by company (Fig 3). Company-specific decreases in total sugar volume (represented by the black marker lines) were predominately due to reductions in the mean sugar content (represented by the orange bars), although some companies also had reduced sales volumes (represented by the grey bars). Where we recorded increases in the total volume of sugars sold, this corresponded to increases in sales volumes for individual companies. Companies that manufacture products in more than 1 category are presented separately in each relevant category. In the case of the companies Mondelez (biscuits), Tesco (breakfast cereals), Thorntons (chocolate confectionery), and Haribo and Mondelez (sugar confectionery), there were also increases in

**Table 2. Data points in the nutrient composition and sales datasets 2015–2018.**

| Measure | 2015 | 2016 | 2017 | 2018 |
|---|---|---|---|---|
| Number of individual products in nutrient composition dataset | 2,515 | 2,535 | 2,443 | 2,351 |
| Number of brands in sales dataset | 350 | 349 | 355 | 353 |
| Number of companies | 95 | 99 | 98 | 97 |

**Table 3. Total volume sales of food category in tonnes, 2015–2018.**

| Category | Volume sales (tonnes) | | | | Percent change (2015–2018) |
|---|---|---|---|---|---|
| | **2015** | **2016** | **2017** | **2018** | |
| Biscuits and cereal bars | 383,600 | 387,500 | 392,500 | 395,400 | 3% |
| Breakfast cereals | 394,500 | 391,000 | 384,100 | 383,100 | −3% |
| Chocolate confectionery | 349,900 | 354,400 | 352,400 | 353,600 | 1% |
| Sugar confectionery | 128,000 | 127800 | 126,800 | 127,400 | 0% |
| Yoghurts | 511,600 | 514,900 | 513,600 | 503,200 | −2% |
| **Total** | **1,767,600** | **1,775,600** | **1,769,400** | **1,762,700** | **0%** |

the mean sugar content. These were attributable to an increased proportion of sales from brands within these companies with a higher sugar content.

## Sensitivity analysis results

**Sensitivity analysis 1—heterogeneity of individual products within brands.** Forty-five percent (*n* = 25) of breakfast cereal brands were classified by the researcher as product range brands. These are brands that have a wide range of individual products and flavour variations, and were judged to have no single product leader in terms of sales. As these brands represent a wide range of individual products and flavours, they were excluded from this analysis as it was assumed taking a mean sugar content across the product range would not impact the results. Fifty-five percent (*n* = 30) of brands were classified as brands with a leading individual product, where taking a mean sugar content across all products might have an impact on the overall sales-weighted mean sugar content. When comparing the sugar content of the leading individual product to the average values used in this study, there was an average variation of −8% (95% CI −88% to 30%), representing a difference of 0.5 g/100 g (95% CI −8.3 to 4.1 g/100 g) in sugar content.

**Sensitivity analysis 2—volume represented by 'others'.** We assumed 2 scenarios for assessing what impact excluding the 'others' sales volume had on the overall sales-weighted mean sugar content of foods, as this volume represented between 7% and 21% of sales depending on the category. Scenario 1 assumed that the brands represented by the 'others' sales volume did not change their sugar content over time. Scenario 2 assumed that the sales-weighted sugar content of 'others' brands changed at the same rate as the overall category. The results of both scenarios were similar to the results presented in this study (Table 6).

**Table 4. The sales-weighted mean sugar content of food categories, 2015–2018.**

| Category | Sales-weighted mean sugar content (g/100 g) (SD, 95% CI) | | Absolute (g/100 g) and percentage change (95% CI) 2015–2018 | *p*-Value |
|---|---|---|---|---|
| | **2015** | **2018** | | |
| Biscuits and cereal bars | 30.0 (9.0, 29.3–30.7) | 28.1 (9.9, 27.3–29.0) | −1.9, −6.3% (−10.0%, −2.7%) | 0.78 |
| Breakfast cereals | 18.8 (9.3, 18.0–19.5) | 16.3 (8.6, 15.5–16.9) | −2.5, −13.3% (−19.2%, −7.4%) | 0.16 |
| Chocolate confectionery | 51.7 (10.6, 50.8–52.7) | 51.2 (10.8, 50.2–52.4) | −0.5, −1.0% (−3.1%, −1.2%) | 0.91 |
| Sugar confectionery | 62.2 (50.0–75.6)* | 60.7 (52.8–69.0)* | −1.5, −2.4% (−4.2%, −0.6%) | 0.92 |
| Yoghurts | 11.2 (4.1, 10.9–11.5) | 9.3 (4.0, 9.1–9.6) | −1.9, −17.0% (−26.8%, −7.1%) | 0.70 |
| **Total** | **28.7 (8.3, 27.2–30.4)** | **27.2 (8.3, 25.8–28.4)** | **−1.5, −5.2% (−9.1%, −1.4%)** | **0.52** |

*Interquartile range given for sugar confectionery due to data not being normally distributed.

**Table 5. Total volume of sugars sold by food category in per capita per day terms, 2015–2018.**

| Category | Total sugar sales, grams/person/day (95% CI) | | Absolute (grams/person/day) and percentage change 2015–2018 |
|---|---|---|---|
| | 2015 | 2018 | |
| Biscuits and cereal bars | 4.9 (4.8–5.0) | 4.6 (4.5–4.7) | −0.3, −6.1% |
| Breakfast cereals | 3.1 (3.0–3.2) | 2.6 (2.5–2.7) | −0.5, −16.1% |
| Chocolate confectionery | 7.6 (7.5–7.7) | 7.5 (7.4–7.7) | −0.1, −1.3% |
| Sugar confectionery | 3.4 (2.7–4.1)* | 3.2 (2.8–3.6)* | −0.2, −5.9% |
| Yoghurts | 2.4 (2.3–2.5) | 1.9 (1.9–2.0) | −0.5, −20.8% |
| **Total** | **21.4 (20.3–22.7)** | **19.8 (18.8–20.7)** | **−1.6, −7.5%** |

*Interquartile range given for sugar confectionery due to data not being normally distributed.

## Discussion

Nutrient composition and food sales data were combined to calculate the sales-weighted mean sugar content of 5 food categories between 2015 and 2018. The results were compared to PHE's 5% and 20% sugar reduction targets for 2018 and 2020, respectively. Our results show there was a decrease in the total volume sales of sugars of 7.5% between 2015 and 2018, of which 70% was attributable to a decrease in the sugar content of foods and 30% to changes in sales volume of specific brands and products. The reduction in the sales-weighted mean sugar content of the selected foods was 1.5 g/100 g, equivalent to 5.2%, which was in line with the PHE interim 2018 target for sugar reduction. There was,

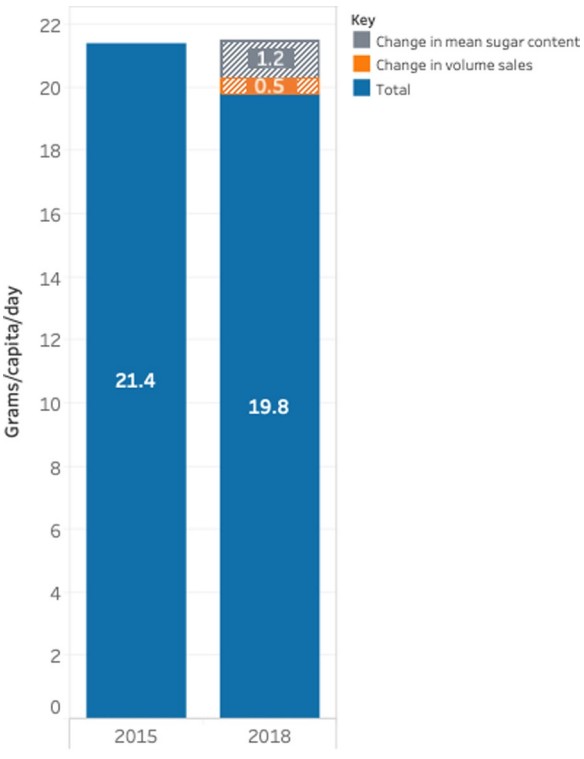

**Fig 2. Change in the total volume of sugars from all food categories combined, split by change in sales and change in mean sugar content, 2015 and 2018.**

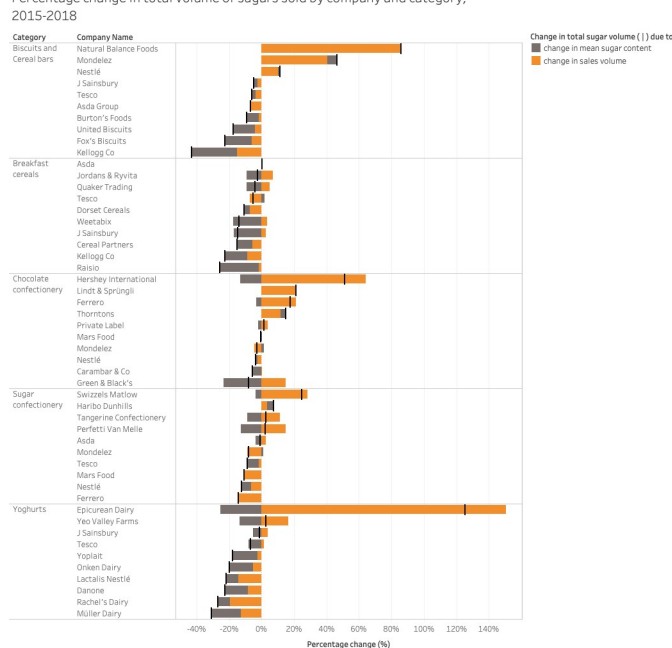

**Fig 3. Percentage change in the total volume of sugar sold by the top 10 companies in each category between 2015 and 2018.** Black marker lines represent the percentage change of total volume of sugars sold by company and category. This total change is split into the percentage change due to changes in volume sales of products (orange) and the percentage change due to changes in the mean sugar content of products (grey).

however, great heterogeneity between categories and companies. The yoghurt and breakfast cereal categories saw the largest reductions in sales-weighted mean sugar content, 17.0% and 13.3%, respectively. In contrast, chocolate confectionery and sugar confectionery saw little change in either the sales-weighted sugar content of products or per capita volume of sugar sold. Of the 50 companies representing the top 10 companies in each category, 24 (48%) met the 2018 category-specific 5% reduction targets. In addition, 4 companies had already met the 20% target for reductions by 2020. There were increases in the sales-weighted mean sugar content of products sold by 10 companies, due to an increase in sales of brands with higher sugar content within those companies.

**Table 6. Main analysis results of sales-weighted mean sugar content of foods in 2015 and 2018 compared to the results of 2 sensitivity scenarios.**

| Year | Overall sales-weighted mean sugar content of foods (g/100 g) | | |
|---|---|---|---|
| | Main analysis[1] | Scenario 1[2] | Scenario 2[3] |
| 2015 | 28.7 | 29.1 | 29.1 |
| 2018 | 27.2 | 27.8 | 27.6 |

[1]The overall sales-weighted mean sugar content of foods presented in the main analysis in this study for 2015 and 2018.

[2]Scenario 1 assumed that the brands represented by 'others' did not change their sugar content over time, and the sugar content remained the same as in 2015.

[3]Scenario 2 assumed that the sales-weighted sugar content of brands represented by 'others' changed at the same rate over time as the overall category.

## Comparison with other studies

To our knowledge, this is the first peer-reviewed study to report on the changes made by individual companies towards reducing the sugar content of foods in the UK. The findings are similar to the few studies that have looked at the sugar content of individual food categories. One study looking at the sugar content of breakfast cereals in the UK found that the absolute mean sugar content of products, based on data collected from 5 online supermarkets, was 20.8 g/100 g in 2015 [18], compared to our finding of 19.1 g/100 g. Another study reported the sugar content of 921 yoghurts sold from 5 online supermarkets in the UK in 2016 [19]. Results are similar to our findings for drinking yoghurts (9.1 g/100 g compared to 9.0 g/100 g in this study), flavoured yoghurts (12.0 g/100 g versus 11.3 g/100 g), and plain/Greek-style yoghurts (5.0 g/100 g versus 4.3 g/100 g).

PHE has previously published its own analysis of the change in the sales-weighted mean sugar content of different food categories, using sales and nutrient composition data from Kantar Worldpanel [14]. PHE reported that the sales-weighted mean sugar content of foods declined by 2.9% overall between 2015 and 2018, compared to the 5.2% (95% CI −9.1%, −1.4%) decrease reported here. Category-level changes from the PHE report are compared with the results of this study in Table 7. Both studies saw the greatest reductions in sugar content in yoghurts and breakfast cereals, with small changes for chocolate and sugar confectionery.

PHE also reported that the total volume of sugar sold from included categories increased by 2.6% from 2015 to 2018, compared to the reduction of 7.5% reported here [14]. The difference between the results is likely to be due to the different categories included in the 2 analyses, and differences in the datasets used. PHE used sales and nutrient composition data from Kantar Worldpanel for monitoring the sugar content of foods [14]. Kantar sales data are based on the results from a sample of measured weekly household purchases scanned by participating households and report much lower overall sales compared to the total annual sales estimates used here from Euromonitor. This may be due to individuals forgetting to scan products, especially impulse purchases consumed out of the home such as confectionery [20]. Although the absolute values differ, the main patterns were similar, with the greatest reductions in sugar content in yoghurts and breakfast cereals, and little change in chocolate and sugar confectionery.

## Strengths and limitations of this study

By pairing nutrient composition data with food sales data, we were able to analyse the sugar content of what was sold, not just of what products were available. This provides insights into how individual companies have reduced the sugar in their products, potentially as a result of PHE's call to reduce the sales-weighted mean sugar content of their products by 20%. We hope

**Table 7. Percentage change in sales-weighted mean sugar content of different food categories between 2015 and 2018 compared to Public Health England findings [14].**

| Category | Percent change in sales-weighted mean sugar content | |
|---|---|---|
| | Public Health England [14] | This study (95% CI) |
| Biscuits and cereal bars | −0.6% | −6.3% (−10.0%, −2.7%) |
| Breakfast cereals | −8.5% | −13.3% (−19.2%, −7.4%) |
| Chocolate confectionery | −0.3% | −1.0% (−3.1%, 1.2%) |
| Sugar confectionery | +0.6% | −2.4% (−4.2%, −0.6%) |
| Yoghurts | −10.3% | −17.0% (−26.8%, −7.1%) |

that these results provide transparent information for stakeholders to assess how the food industry is progressing towards public health targets.

The nutrient composition data used here were collected by Edge by Ascential from the websites of the UK's 3 leading retailers (Tesco, Sainsbury's, and Asda) on the same date (13 December) in each year included in this study. The total number of products included is therefore likely to be an underestimate, as it is a reflection of what is available online on a single date, and does not include products that are available for purchase from other retailers, independent stores, and markets. Taking data from single time points also means that we have not captured the churn of products that are entering and being removed from the market seasonally and over the course of the year.

This study does not cover the entire food market, nor does it cover every category included in PHE's sugar reduction targets. Cakes, morning goods, ice cream, puddings, and sweet spreads were excluded due to a lack of granularity or lack of alignment between PHE's categories and the Euromonitor sales database. According to the National Diet and Nutrition Survey, these missing categories represent an estimated 11% of sugar intake for children aged 11–18 years [7].

Using sales data—as opposed to dietary survey data to estimate intake or household panel data to monitor purchases—has 2 main advantages. First, it avoids reliance on individual recall of consumption, and underreporting in scan data [20], and, second, sales datasets include granular details about the individual brands that have been sold. Euromonitor has wide coverage, including hypermarkets, supermarkets, convenience stores, vending machines, and fast food outlets. This means that it is particularly suited to studying changes at the company level.

However, there are some major limitations of using Euromonitor sales data. Euromonitor does not cover the whole market, and a lack of granularity in the data meant that some high-sugar categories, including ice cream, cakes, and pastries, were not included in this study. Therefore, limited conclusions can be drawn in terms of how the sugar content of all foods has changed in the UK. Another limitation is that Euromonitor groups small and local brands under the umbrella term 'others', meaning that a proportion of volume sales could not be paired with nutrient composition data. The results of the sensitivity analyses, which examined 2 different scenarios for estimating the sugar content of these products, demonstrated that excluding the 'others' sales volume was unlikely to have had any significant impact on the results. However, it does mean that smaller, local brands are essentially excluded from this study and means the findings are less representative of the whole market.

A more major limitation is that the food sales data were only available at the brand level, not the individual product level, and therefore any heterogeneity that occurred between products under the same brand would been have missed by assuming all products are sold equally. Results of the sensitivity analysis using the breakfast cereal category as an example showed that this is unlikely to have affected around half (45%) of brands, as these represent a broad variety of products. However, for the remaining 55%, the sugar content value used in this study and the sugar content of the leading individual product differed, with an overall percentage difference of −8% (95% CI −88% to 31%), or 0.5 g/100 g (95% CI −8.3 to 4.1 g/100 g). These results suggest that while the study's overall findings are not likely to be affected by assuming all products within a brand are sold equally, there may be some significant misrepresentations at the brand and potentially company level. Use of individual-product-level sales would improve this, although datasets that have product-level sales data have other limitations, including their cost and limitations in publication of company and brand names [21].

Users of third-party sales databases have no control over the data collection process, and there is limited transparency in the methods of data collection, or the reliability of sources [21]. The sales data used in this study are for the total UK population and are not broken

down by any sociodemographic factors; therefore, any variation in the impact sugar reduction might have based on income and brand preference could not be determined in this study. Sales data do not account for waste, but in this time trend analysis, the absolute values are less pertinent than the change, assuming that there have not been major changes in food waste.

## Policy implications

Although our analysis suggests the PHE 5% interim target for 2018 was met overall, the majority of this change was driven by just 2 categories, yoghurts and breakfast cereals, with negligible changes in sugar and chocolate confectionery. The difference between categories may be due to differences in the technical ease of reformulating products. It is important to note that PHE's sugar reduction targets sit alongside long-running public health awareness campaigns, including Change4Life [9], as well as increased attention from the mass media about the sugar content of everyday products, and therefore the results observed are not solely due to PHE's setting targets. The larger reductions observed in the sugar content of yoghurts and breakfast cereals may also reflect pressure from the public health community about the sugar content of these products that are otherwise considered part of a healthy diet, but where the sugar content is perceived to be 'hidden'. In contrast, there has been less media attention paid to the sugar content of confectionery, perhaps because it is seen as an indulgence and the sugar content is more overt.

This analysis raises concerns about the likelihood of achieving the more stretching 20% reduction target set for 2020. Over half of companies included in this analysis had not met the 5% reduction target, and since companies that made the greatest reductions and already achieved the 2020 target may now slow or pause their sugar reduction efforts, there will need to be a considerable acceleration in reductions by other companies, especially in the confectionery categories. It is also important to note that the categories that have seen the greatest changes (breakfast cereals and yoghurts) are also those that had the lowest levels of sugar to start with. Further research using more recent data could be conducted to assess further changes in sugar content between 2018 and 2020.

It is notable that the level of change in the sugar content of foods is smaller than that seen in soft drinks. Using similar methodology, we previously showed that the sales-weighted mean sugar content of soft drinks fell by 34% over the same time period [22], compared to 8% for the food categories studied here. The relative success of sugar reduction in drinks over that in foods may partly be due to the greater technical challenges in reformulation of foods. In drinks the sweetness delivered by sugar and other caloric sweeteners can be replaced with high-intensity sweeteners [23]. Foods containing starches and/or fats combined with sugars are harder to reformulate [23]. In food categories such as those included in this study, sugar not only delivers sweetness but has other technical properties, including water retention, browning, texture modification, and structure [23]. The consumer acceptability of reduced-sugar foods is also thought to be lower than that of their regular counterparts [24], meaning that food companies may be reluctant to reduce the sugar content of their products [24,25], especially for indulgent products such as confectionery and biscuits.

Alternatively, the small changes in the sugar content of foods compared to drinks may reflect differences in the policy context. In the UK, sugar reduction in soft drinks has been driven in part by the Soft Drinks Industry Levy, with high-sugar products (>8 g/100 ml) being subject to a tax of 24 pence per litre and mid-sugar products (5–8 g/100 ml) being taxed at 18 pence per litre [26]. Evidence shows that the introduction of the levy led to companies reformulating their products to less than 5 g sugar per 100 ml in order to avoid the levy, leading to significant reductions in the sugar content [27] and the total volume of sugar sold from soft

drinks [22]. In contrast, the food categories included here are subject only to a voluntary programme of sugar reduction targets monitored by PHE, but with no penalties for lack of progress. Without sanctions, food companies may be less motivated to engage with this voluntary programme.

The PHE sugar reduction targets are designed to help the population achieve dietary recommendations for free sugar intake, which in the UK are no more than 19 g for children aged 4–6 years, 24 g for children aged 7–10 years, and 30 g for those aged 11 years and above [5,6]. Assuming a direct relationship between sales and consumption, this analysis suggests a per capita intake of free sugars of 19.7 g/d from these food categories alone. Given that these categories provide only 40%–60% of total free sugar intake [7], it is clear that much greater reductions in these and other categories of foods and drinks will be required to meet dietary recommendations.

The UK claims to be the first country in the world that has implemented a structured sugar reduction programme with incremental targets [28]. However, other countries have implemented a range of other policies to reduce the availability and affordability of high-sugar foods. For example, in Chile, high-sugar products have had to display a 'high in sugar' front-of-package warning label since June 2016 [29]. A prospective study has shown that the proportion of breakfast cereal products that are classified as high sugar fell from 46% in 2015–2016 to 24% in 2017, with little change in sweet confectionery. In Mexico in 2013, a tax on nonessential energy-dense foods that contain >275 kcal/100 g was implemented, alongside a nationwide public health campaign. An initial evaluation showed that purchases of these products fell by 5% 2 years after the implementation [30]. Hungary introduced a tax in 2011 on a range of processed foods based on their sugar content, including confectionery and snacks. An initial evaluation of this policy showed that prices of tax-eligible products increased by 29% and consumption declined by 3% [31]. These case studies may provide examples of other actions that could be taken in the UK to accelerate progress on sugar reduction.

In conclusion, our findings suggest there has been a mixed response by companies to reducing the sugar content of foods in the UK between 2015 and 2018. The greatest reductions in sales-weighted mean sugar content were observed in breakfast cereals and yoghurts, with minimal change in sugar and chocolate confectionery. The majority of companies had not met the 5% sugar reduction target by 2018, suggesting that additional policy measures may be needed to further reduce the sugar content of foods.

## Supporting information

**S1 STROBE Checklist.**
(DOCX)

## Author Contributions

**Conceptualization:** Lauren K. Bandy, Peter Scarborough, Mike Rayner, Susan A. Jebb.

**Data curation:** Lauren K. Bandy, Peter Scarborough.

**Formal analysis:** Lauren K. Bandy, Peter Scarborough.

**Investigation:** Lauren K. Bandy.

**Methodology:** Lauren K. Bandy, Peter Scarborough, Richard A. Harrington.

**Project administration:** Lauren K. Bandy.

**Resources:** Lauren K. Bandy.

**Supervision:** Peter Scarborough, Richard A. Harrington, Mike Rayner, Susan A. Jebb.

**Visualization:** Lauren K. Bandy, Peter Scarborough.

**Writing – original draft:** Lauren K. Bandy, Susan A. Jebb.

**Writing – review & editing:** Lauren K. Bandy, Peter Scarborough, Richard A. Harrington, Mike Rayner, Susan A. Jebb.

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
