## [Editor Report · Decision Letter 0]

20 Jul 2020

Dear Dr Bandy, 

Thank you for submitting your manuscript entitled "Assessing progress by UK companies to reduce total volume sales of sugar in selected food categories" for consideration by PLOS Medicine.

Your manuscript has now been evaluated by the PLOS Medicine editorial staff [as well as by an academic editor with relevant expertise] and I am writing to let you know that we would like to send your submission out for external peer review.

Kind regards,

Adya Misra, PhD,

Senior Editor

PLOS Medicine

---

## [Decision Letter · Decision Letter 1]

15 Oct 2020

Dear Dr. Bandy,

Thank you very much for submitting your manuscript "Assessing progress by UK companies to reduce total volume sales of sugar in selected food categories" (PMEDICINE-D-20-03288R1) for consideration at PLOS Medicine. 

[LINK]

In light of these reviews, I am afraid that we will not be able to accept the manuscript for publication in the journal in its current form, but we would like to consider a revised version that addresses the reviewers' and editors' comments. Obviously we cannot make any decision about publication until we have seen the revised manuscript and your response, and we plan to seek re-review by one or more of the reviewers. 

We expect to receive your revised manuscript by Nov 05 2020 11:59PM. Please email us (plosmedicine@plos.org) if you have any questions or concerns.

We look forward to receiving your revised manuscript. 

Sincerely,

Adya Misra, PhD

Senior Editor 

PLOS Medicine

plosmedicine.org

Notes from the Academic Editor

Please clearly state the weaknesses and flaws in the nutrient data base collected. In particular in the weights used to give all foods the same weights. This potentially means that obscure foods in one category that are truly niche foods would have the same weight as a popular food item [e.g. a product with no reformulation].

By using weights of sales instead of longitudinally following specific products, this is not true reformulation at all but rather an attempt to examine sugar reductions. This can only be done by looking at each product. Therefore, weighted by product and not by category with the same weights for all. These inherent limitations must be discussed clearly in a revision. 

Please revise your title according to PLOS Medicine's style. Your title must be nondeclarative and not a question. It should begin with main concept if possible. "Effect of" should be used only if causality can be inferred, i.e., for an RCT. Please place the study design ("A randomized controlled trial," "A retrospective study," "A modelling study," etc.) in the subtitle (ie, after a colon).

Abstract

Abstract- please replace “similar period” with the exact timeline 

Last sentence of the methods and findings must outline 2-3 limitations of your study design/methodology

Please add exact p-values, unless p<0.001 as needed

Please temper all results and conclusions by adding “our results show ..” or similar. 

Conclusions * Please address the study implications without overreaching what can be concluded from the data; the phrase "In this study, we observed ..." may be useful. * Please interpret the study based on the results presented in the abstract, emphasizing what is new without overstating your conclusions. * Please avoid vague statements such as "these results have major implications for policy/clinical care". Mention only specific implications substantiated by the results.

Please use square brackets for references and Vancouver style for bibliography 

Discussion

Please begin this section with 1-2 sentences summarising what was done

Several sentences need toning down, for example “Most progress was made in the yoghurt and breakfast cereal categories ..”. I suggest revising to “our results show most sugar reductions were observed in xxx categories” or similar 

Line 191- I’m not sure consumer behaviour can be inferred from this data, at least not without major caveats so I suggest removing

Please avoid assertions of primacy and add “to our knowledge” at line 221

Throughout- while you present the findings as a direct result of PHE voluntary targets, there is no reason to believe that these changes in sugar content were independent of the PHE. I suggest de-emphasizing this aspect of your manuscript and noting that the results “could be” due to PHE directives. 

 Please ensure that the study is reported according to the [STROBE] guideline, and include the completed [STROBE or other] checklist as Supporting Information. When completing the checklist, please use section and paragraph numbers, rather than page numbers. Please add the following statement, or similar, to the Methods: "This study is reported as per the Strengthening the Reporting of Observational Studies in Epidemiology (STROBE) guideline (S1 Checklist)."

Did your study have a prospective protocol or analysis plan? Please state this (either way) early in the Methods section.

Comments from the reviewers:

Reviewer #1: The study by Brandy LK assessed the progress made by UK food companies in sugar reduction in five food categories covered by PHE's voluntary sugar targets (biscuits and cereal bars, breakfast cereals, chocolate confectionery, sugar confectionery and yoghurts) using public available sales data and nutrient composition information of 2515 products under 250 brands nested in 95 companies. They found a 7.5% reduction in total volume of sugars sold per capita and 5.2% in sales-weighted mean sugar content. The sugar reduction was primarily observed in yogurts and breakfast cereals, and more than half the companies did not meet the PHE's 2018 target. This study provided the latest evidence on how the UK food industry responded to PHE's voluntary sugar reduction program and highlighted the issue of accelerating the sugar reduction progress in categories such as confectionery for most UK companies. 

The paper is well written. However, there are several points that need further clarification: 

1. Can you explain how you generated the percentage (line 152-153) and absolute value (Figure 1) that "reduction in the mean sugar content" or "decrease in volume sales" contributed to the total sales-weighted volume of sugar sold? 

2. This study calculated the mean sugar content for each brand by simply averaging out the sugar content of all the product variants (line 98-101). However, the sales might differ greatly among product variants within the same brand (for example, the regular chocolate spread might be more popular than sugar-reduced ones), thus we suggested that the mean sugar content of each brand should be weighted by product-level sales if data available. 

3. Does the Euromonitor covers the sales data of the five food categories in the UK? Can you provide more information about the coverage of Euromonitor sales data? 

4. The number and the subcategories listed in table 1 for "sugar confectionery"? are not consistent. Please double check.

5. Please double check the percentage figure of total volume sales for the 62 and 42 brands that can't be matched for their nutrient composite data (line 103 and 106). Should it be 3% in line 103 and 2% in line 106?

6. Where does the percentage "five categories which together provide 40-60% of free sugar intake in the UK" (line 175) come from? Can you provide a reference here?

7. Can you explain more about the potential reason why the %change in sales-weighted mean sugar content between 2015 to 2018 was much smaller in PHE's findings compared to this study? Why the sales weighted sugar content of sugar confectionery category showed a reduction (2.4% [-0.6%, -4.2%]) in this study, whereas there was a small increase (+0.6%) in PHE's report?

Reviewer #2: Comments

Generally, I think this is an interesting paper that provides a useful contribution to public health. Reformulation of the food supply is a topic of major interest with regards to population-level strategies to improve diets, prevent obesity, and related NCDs, and in particular it is of great interest whether voluntary pledges vs. mandatory regulations are needed to improve the nutritional quality of the food supply. This paper focused in the UK is useful, especially in comparing and contrasting the changes that have been observed in response to this voluntary initiative compared to UK's mandatory sugary drink tax. 

Although I think my comments below are readily addressable, I did find the overall quality of the paper difficult to assess due to some confusion in the methods section. I think that adding some more detail on what exactly was done, as well as the rationale, would help tremendously. I look forward to reading future revisions.

Major queries:

* Clarity around brand-company-product. It was at times difficult to follow what you meant by each term, and they seemed to be used interchangeably in some cases.

* Clarity around sales weighting: providing rationale for why to do the sales-weighting and how exactly it was done (seemingly at the brand level). It could be useful to provide an example. 

* Clarity around the sample: did products move in and out? Or you followed specific products over time?

* Additional consideration and potentially sensitivity analyses around a) the implications of missing data (e.g. missing categories, other missing data); but also the key limitation that you are weighting at the brand level, which doesn't incorporate the considerable heterogeneity that likely occurs for products within brands. Even if you could do a sensitivity analysis of one specific brand and products within it to show how much variation you might expect to see if one product happened to be sold much more than another product, that would be helpful. 

* This article is framed as a pre/post study to address the question, did companies respond to PHE's initiative, but it's impossible to know whether they were already reformulating their products and PHE's initiative just happened to coincide with it. Have you considered adding additional years prior to PHE's announcement? 

Intro

It would be useful to link sugar to health, and particular free sugar, as there is a slight but important difference between the concept of free sugar and total sugars, which it seems like PHE is targeting. 

you might explain briefly what is PHE for non-UK readers. It would be very relevant to know if this is a government org, ngo, industry-backed group, etc. 

Consider adding more rationale about why we care if this voluntarily initiative works or not? Particularly in the 2nd to last paragraph, why is it important to understand variability between companies? 

How does this link to global efforts (mandatory or voluntary policy actions) to reduce sugar?

Consider more rationale about why it is important to consider sales-weighted data (vs. simply looking at changes in the nutrient composition of products)

When did PHE actually release their guidelines? It would be useful to understand to provide context for time window that was used in this study. 

Methods

Please state what were the other 4 categories you could not analyze? Do you know what % of sales these categories account for? This is an important point, particularly as I noted that beverages were not included at all in this analysis. Was that b/c of Euromonitor or because they weren't included in PHE's initiative? This is important considering all the other sugar reduction efforts going on in beverages. Also, if this does focus on foods, it would be useful to strengthen in the intro why it is important to consider foods in the diet, not just sugary drinks. 

In your explanation of the brand, it's not totally clear what you're saying. It sounds like Cadbury Dairy Milk and Oreos are considered two separate brands, right? 

Why only consider such a small set of years (2015-2018)? Isn't it useful to consider additional years prior to PHE's announcement to understand if trends accelerated?

With regards to understanding reformulation, did you include products in the analysis that could not be reformulated because they did not contain added sugar originally (e.g. plain oatmeal)?

Please be clear if Euromonitor was providing brand-level or product-level data. It sonds like it was at the brand level. How did you identify all the product within the brand? Secondly, you note that the data is "sales-weighted," but it is sales weighted at the brand level, right? So if there is variability within the brand by individual product, you won't see that here. 

How frequently is the nutrient composition data updated?

I got confused between brands, products, and companies. For example, in Line 102, you note you couldn't find nutrient data for 62 brands- but in many cases a brand does not equal a product, which is what the nutrient data is on?

It would be helpful to understand the total scope of missing data, at least in the discussion. Sounds like you have about ~17% volume sales missing from missing nutrient data, plus the 4 other categories you could not find in Euromonitor. Would you get a different picture if purchases data were used instead? 

Analysis: would be useful to understand if brand and company are synonymous. I was a little surprised to see this reported by company when previously you were talking mainly about brands. Also, I'm assuming brands are followed longitudinally over time? Or are these repeated cross sections? What is the sample size. 

I didn't understand the structure of the data. Was it at the product-level, with covariates that included year, category, and company?

Please explain how the sales-weighting by brand level was done in the context of a dataset on products.

Results:

Please include units in table 3

Please include some measure of statistical significance in the text when you describe results (Line 140)

Line 140- the absolute reduction in biscuits was quite similar to that ovserved in yogurt and cereal, which had higher relative declines mainly because they started out at lower amounts. You might just note this by highlighting the absolute vs. relative change. 

How did you conduct the analysis that gave rise to the result about product-level changes vs. sales-level changes reported in Line 152-153?

Figure 1 is confusing. The bars look as though they represent the total sugar sales in each year, but then the stacked bar in 2018 represents change? I think you would want to find some way to represent that the 1.2 and 0.5 represent sugar that is no longer in the food supply (whereas currently it looks like part of a bar that represents sugar currently observed in the food supply). 

On the company analysis, did some companies product products across categories? If so, how did you handle this in analysis?

Line 164- when parsing out the change due to sugar content and change due to sales, it's not clear why you would want the sugar content to be sales weighted. Isn't the point to show specifically what is going on at the product level, not accounting for changes in behavior (sales)? This is where it is also useful to understand if you only included products that remained in the data over time, since it is unclear how new product entry or product exit affected these numbers (a different concept from reformulation, which is changing the nutrient content of a specific product).

Figure 2: I found it hard to understand whether within each company it has a net increase or decrease in sugar sales. This could benefit from a sentence or two to help the reader interpret what is happening in this figure. 

Worth noting in the discussion that several of the categories that changed most had the smallest levels of starting sugar levels (and particularly in yogurt, some of this is natural sugar). 

Maybe not surprising not to see a reduction in confectionary. Could you discuss how the changes observed might also reflect demand? It seems like if someone is making the choice to have a candy bar, they don't necessarily want it to have reduced sugar levels, whereas this could be a more desired attribute for seemingly healthy products like cereal or yogurt, or even for products that have more variability in sweetness, such as biscuits/cookies.

Discussion

Were you able to look at non-sugar sweeteners? (Is this mandatorily reported in the UK)? It woudl be useful to know if these sugar reductions resulted in a reduction in overall sweetness of the products, or if they were just being replaced with NNS. I would also look more into UK/global trends in NNS. My understanding is that NNS-containing foods are increasing, not just for drinks but in foods as well (in fact, yogurt is a popular category for this). 

The paragraph stating line 294- I think it would be more useful to compare to countries that expicitly addressed sugar as part of their policy. In particular, consider the countries that are explicitly including a warning on high-sugar foods. Those are currently: Chile, Peru, Israel; currently being implemented in Mexico and Uruguay. Chile has some results, including a paper on reformulation that was published in Plos Med (Reyes et al)- this may actually be the most relevant comparison for this paper.

Limitations section- it would be useful to discuss more the implications of missing data and the fact that you had to (I think) weight all products within a brand equally. For example, what if one product within a brand accounts for most of the sales in that brand? Like if there was a low-sugar Twix that was developed, for example, but everyone continued purchasing the main Twix, this would show a misleading reduction in sugar for this brand. 

Also talk about how well you can say that these changes were attributable to PHE's target. You didn't use a counterfactual here. Is 

Also consider discussing limitations for what we can understand on public health impact from this study. For example, the effect of sugar reformulation could be greater or lesser for different socio-demographic populations, depending on what brands (and products within the brand), they buy.

Reviewer #3: 

This review relates to manuscript PMEDICINE-D-20-03288R1 titled "Assessing progress by UK companies to reduce total volume sales of sugar in selected". The manuscript is well-written and easy to follow. 

My main concern relates to the fact that about 20% of data was excluded from the analysis (cf. lines 102-109) without any sensitivity analysis conducted to assess the potential impact of missing data on the results. I would encourage the authors to consider additional analyses under various assumptions about the missing data.

Minor comments

* On line 33, the abstract mentions 99 different manufacturers but online 42, "50 companies" are mentioned (line 42). Then at the start of the results section (line 128) 95 companies are mentioned. Please clarify/correct as relevant.

* It is not always clear which metric is the primary outcome. Given that targets were set for mean sales weighted sugar content, it seems that this should be the primary outcome

* Please clarify whether all existing sub-categories were included in the 5 categories analysed. For example: 7 sub-categories were included under the "biscuits and cereal bars" category. Are these 7 sub-categories all existing categories under "biscuits and cereal bars"?

* Please provide additional details about the calculation of "sales-weighted mean sugar content" explaining the rationale for weighting the data and the details of the calculation.

* Given that the same brands/products were analysed in 2015 and 2018, did the analysis consider the lack of independence ("pairing") between the two periods?

* Please clarify whether the unit of analysis was the product or the brand.

* Please explain in more details the rationale for weighting the data so that each brand is proportional to total sales and having the sum of the unit of analysis equal to the number of brands. 

* Estimates are presented with confidence intervals as per usual practice; however, it strikes me that the data presented represents the entire "population" of products in the database and is therefore not subject to sampling uncertainty. I am not sure that confidence intervals are particularly meaningful in this case. To give some idea of the variability, I would suggest instead presenting standard deviations and/or quartiles.

* I suspect that changes in mean sugar content were only calculated for products present in both 2015 and 2018. Please clarify.

* Please consider displaying histograms of % change in sugar content by product for products present in 2015 and 2018.

* Please consider displaying all 5 categories on Figure 1. In doing so, one could show only the 2018 bar per category since it shows both the 2015 and 2018 levels together with where the change occurred (mean sugar content vs volume sales).

* Figure 2 is a nice way to summarize changes by company. I believe additional text would however help the reader understand the Figure. A complementary figure could be a histogram of Percentage change in the total volume of sugar sold for all companies (without indentifying companies)

-Laurent Billot

[LINK]

---

## [Decision Letter · Decision Letter 2]

10 Mar 2021

Dear Dr. Bandy,

Thank you very much for submitting your revised manuscript "The sugar content of foods in the UK by category and company: A repeated cross-sectional study, 2015-2018" (PMEDICINE-D-20-03288R2) for consideration at PLOS Medicine. 

I apologize for the delay in getting back to you. Your paper was evaluated by a senior editor and discussed among all the editors here. It was also discussed with an academic editor with relevant expertise, and sent to two of the original reviewers, including a statistical reviewer. The reviews are appended at the bottom of this email and any accompanying reviewer attachments can be seen via the link below:

[LINK]

In light of these reviews, we would like to consider an additional revised version that addresses the reviewers' and editors' comments. Obviously we cannot make any decision about publication until we have seen the revised manuscript and your response, and we may seek re-review by one or more of the reviewers. 

We expect to receive your revised manuscript by Mar 31 2021 11:59PM. Please email us (plosmedicine@plos.org) if you have any questions or concerns.

We look forward to receiving your revised manuscript. 

Sincerely,

Caitlin Moyer, Ph.D.

Associate Editor 

PLOS Medicine

plosmedicine.org

1.Please completely address the comments from the two reviewers.

2. Data availability statement: Thank you for the links to the Euromonitor and Edge by Ascential webpages. For each data source used in your study: 

3. Abstract: Line 27: Please revise “progress made by UK companies” as it doesn’t seem to reflect the observational nature of the study, and could be rephrased as, “The aim of this study was to assess the sales-weighted mean sugar content and total volume sales of sugar in selected food categories among UK companies between 2015 and 2018” or similar.

4. Abstract: Methods and Findings: Please clarify brand, company, and manufacturer- if any of these terms indicate the same thing, please use consistent terms here and throughout.

5. Abstract: Methods and Findings: Please clearly emphasize that you are noting the key limitations of the study in the final sentence: “The main limitations are that this study does not encompass…”

6. Author summary: Under “Why was this study done?” please revise the third bullet point to make it clear that the study is not directly examining and evaluating progress attributable to the targets.

7. Author summary: What did the researchers do and find? It might be helpful to mention the sugar content measure was weighted by sales volume.

8. Author summary: What do these findings mean? Please remove “...but it’s unlikely the 20% sugar reduction target will be reached by 2020.”

9. Methods: Analysis plan: Thank you for clarifying the study had no pre-planned analysis protocol. However, please indicate all changes to the analysis after the data were examined, such as those arising from peer-review (such as adding sensitivity analyses).

10. Results: Line 194-195: Please clarify what is meant by “top 10” (i.e. in size, in sales volume, etc) in the sentence “...although for clarity, only the top 10 companies for each category are presented in the figures”

11. Discussion: Strengths and Limitations section: The results of sensitivity analyses presented here should be presented in detail in the Results section. More detail would be helpful in classifying the “product range” vs “one product leader” brands, and in addition to the 8% average variation in sugar content for the “leader” product and brand as a whole, the ranges should be presented. Details of how the sensitivity analyses were done should also be included in the Methods.

12. Discussion: Please re-organize the Discussion as follows: a short, clear summary of the article's findings; what the study adds to existing research and where and why the results may differ from previous research; strengths and limitations of the study; implications and next steps for research, clinical practice, and/or public policy; one-paragraph conclusion.

13. Discussion: Line 412-413: Please revise to avoid any implication of a causal relationship between your observations and the PHE targets: “In conclusion, our findings suggest there has been a mixed response by companies to the PHE voluntary sugar reduction targets.”

14. Discussion: Line 415-417: Please revise as the study results do not inform on 2020 data. “Our results show that it is unlikely that the selected categories will meet PHE’s 20% reduction targets by 2020…”

15. Table 3: Please indicate that % change is a comparison between 2015 and 2018

16. Table 4: Please indicate if there is a negative sign missing under percentage change for chocolate confectionery (1.2% vs -1.2%)

17. Figure 2: Please indicate in the title or x axis label that this reflects 2015 vs 2018.

18. References: Please use the "Vancouver" style for reference formatting, and see our website for other reference guidelines: https://journals.plos.org/plosmedicine/s/submission-guidelines#loc-references

19. Checklist: Thank you for including the STROBE checklist. Please revise the checklist, using section and paragraph numbers rather than page and line numbers to indicate locations in text.

Comments from the reviewers:

Reviewer #2: The authors have improved the presentation of results in response to reviewer comments. They are clear about the limitations of using brand-level data in this paper. While this lack of granular product-level data presents limitations, ultimately I do think that because of the challenges and expense of obtaining product-level sales data, with additional clarity in the methods section, this paper could serve as an example to other researchers wishing to analyze category/brand level data using Euromonitor. In addition, this is an interesting and useful paper to understand the extent of sugar reductions in the UK in recent years. A couple of outstanding comments remain:

In my view, the sensitivity analyses are helpful but incomplete. For example- one reviewer pointed out the concern w/missing data (uup to 21.8% for cereal bars). To address this, authors just imputed the same reduction changes as the rest of the category. However, given that these are small and local companies, there is no reason to assume that they would be as able as larger companies to reduce their sugar. A conservative sensitivity analyses would at least assume zero sugar reduction in this group. In addition, it's not clear why you did the sensitivity analyses only for the "umbrella/other" brands and not also the 42 brands for which you couldn't find nutritional data. I don't think simply assuming that the missing brands were the same as included brands is very helpful. 

Why not also present non-weighted data at the product level? This would be a useful sensitivity check that would allow you to avoid the problem that you can't weight by sales at the product level. Then you can come up with a plausible range of estimates for how much sugar reduction occurred.

Finally, the methods section is clearer, but still could use a bit more work (see detailed suggestions below). I still also did not understand entirely how you decomposed the total changes into sugar reductions vs. behavioral changes (sales), since my understanding is those two components have different units of measure. More detail and perhaps an example could help here as well. This also might benefit from creating a specific "outcomes" section which explains in more detail how outcomes are operationalized. 

Minor:

Abstract

Line 32 Note that these are linked at the brand level

Line 41 Please state earlier what the analysis on companies was and if this was different than from the brand-level analysis.

Methods:

In absence of a pre-registered protocol, you should note any analytic changes. These would include sensitivity analyses requested by reviewers, so you can just note that. 

Line 138 what does it mean that the brands were "grouped" by company? Simply that you had an additional variable that was company, or products were at the company level, or that you adjusted your SEs for correlation within companies?

The description of products, brands, and companies has improved, and the example in Line 152 is helpful. However, additional clarity is needed to help the reader follow the line of thought. In part, this is because information about how the dataset was created is spread across the methods and data anlaysis section. I recommend starting out with the most disaggregated form of data (product) and then talk step by step how you arrive to the category level. I would keep the data analysis sectoin purely for the actual analysis. Prior to this, you could also create an "outcomes" section that provides detail on the outcomes you used and why (e.g., why sugar vol, the per capita outcome, the sugar-reduction targets).

For example, in Line 147 it is confusing how you mention sales-weighting right away. I think you need to make it explicit here that brands were matched to products at the brand level. Within brand, all unique product types contributed equally (were averaged?). Here is where I would mention that products could fall in or out of the sample, depending on their availability in the market in the respective year (it is confusing to mention this not until Line 174).

And then note that, within a food category, the *brand* level data were weighted by sales. 

Finally, it is not clear how the analyses were done incorporating both brands and companies. Did you weight brands within companies, as well as in categories?

You could consider a flow chart as well to help readers visualize this. 

Line 175- how did you use ONS stats to calculate per capita sales? did you just divide sales by # of people?

Line 180- it still is not clear to me how you split the change into mean change in sugar content vs. change in total volume sales. In your formula of total sales, what is the outcome? Expenditures, in dollars or sales? From the figure, it looks like it is in grams (of product? of sugar?). I don't understand how you can add sugar content (presumably in grams of sugar) to change in volume. 

Sensitivity analyses should be reported in the results section.

Reviewer #3: My previous comments have been adequately addressed. I am just not sure that the sensitivity analysis conducted to account for missing data is sufficient. The authors have assumed the same reduction in sugar content in the 'other' category than overall. I wonder whether a more conservative assumption should be tested as well e.g. by assuming a smaller change in the 'other' category. Otherwise, I would like the authors to provide additional justification supporting the assumption of a similar reduction.

-Laurent Billot

[LINK]

---

## [Decision Letter · Decision Letter 3]

28 Apr 2021

Dear Dr. Bandy,

Thank you very much for re-submitting your manuscript "The sugar content of foods in the UK by category and company: A repeated cross-sectional study, 2015-2018" (PMEDICINE-D-20-03288R3) for review by PLOS Medicine.

I have discussed the paper with my colleagues and the academic editor and it was also seen again by two reviewers. I am pleased to say that provided the remaining editorial and production issues are dealt with we are planning to accept the paper for publication in the journal.

[LINK]

We look forward to receiving the revised manuscript by May 05 2021 11:59PM.   

Sincerely,

Caitlin Moyer, Ph.D.

Associate Editor 

PLOS Medicine

plosmedicine.org

Requests from Editors:

1. Data availability statement: For clarity, please remove “from the authors” from the sentence: “Due to licencing restrictions, the Euromonitor and Edge by Ascential datasets can only be requested from the authors under licence for the purpose of verification and replication of study’s findings via the research group’s Data Access Committee (contact: Trisha Gordon

trisha.gordon@ndph.ox.ac.uk).” as it seems that the request would go to the Data Access Committee contact person (importantly, the contact for data access request cannot be the authors of the study).

2. Abstract: Methods and Findings: Please briefly describe how sales information in Euromonitor represents all sales in the UK (a fraction of the total sales).

3. Abstract: Methods and Findings: Please mention the total number of companies here: “Of the companies presented in this analysis…”

4. Author summary: Why was this study done? First bullet point: Please change “intakes” to “intake” in the first point.

5. Author summary: What do these findings mean? In the first point, it seems as if this could also be described as “approximately half of the companies” although technically a majority. “...and the majority of companies had not met the 5% sugar reduction target by 2018.”

6. Throughout: For in-text citations, please use square brackets. For multiple references, please do not use spaces within brackets, for example [12,13].

7. Methods: Line 138-139: Please briefly describe the fraction of total sales covered in Euromonitor.

8. Methods: Lines 195-196: Please indicate where these results are shown. Please describe results with p values (for example, in Figure 2 or Table 5). “Chi-squared tests were used to test for differences between the mean sugar content of categories in 2018 compared to 2015.”

9. Results: Line 276: Please briefly mention again the definition for ‘product range’ brands.

10. Results: Line 284: For the second sensitivity analysis, please briefly mention again the significance of the volume represented as “other” in the analysis. Please briefly mention the two scenarios.

11. Discussion: Limitations: Line 364: In the paragraph describing Euromonitor data, please mention that Euromonitor does not cover all sales and the implications of this.

12. References: Please check the formatting of each reference. For example, information is missing from #2, 3, and 4. Please check journal abbreviations (for example, PLoS Med in Reference 29). Please use the "Vancouver" style for reference formatting, and see our website for other reference guidelines https://journals.plos.org/plosmedicine/s/submission-guidelines#loc-references

13. Table 6: Please provide a legend, with a description that clearly describes what is shown in the table. The table should be able to be interpreted on its own.

14. Table 7: It may be helpful to include the reference number/citation for this study in the table or legend.

15. Figure 3: Please include a descriptive legend for this figure.

Comments from Reviewers:

Reviewer #2: Nice job on incorporating the changes. In particular, I really liked the flow chart- this could be used in future studies of nutrition and sales/purchases data. My only comment is that your very final bubble in the flow chart could include what the outcome was (for example, a mean of means). 

Reviewer #3: I have no further comments.

-Laurent Billot

[LINK]

---

## [Editor Report · Decision Letter 4]

7 May 2021

Dear Dr Bandy, 

On behalf of my colleagues and the Academic Editor, Barry M. Popkin, I am pleased to inform you that we have agreed to publish your manuscript "The sugar content of foods in the UK by category and company: A repeated cross-sectional study, 2015-2018" (PMEDICINE-D-20-03288R4) in PLOS Medicine.

Please also make the following two changes to the text:

- When formatting in-text references for multiple references, please include all references within a single set of brackets, but do not use spaces between references, for example [12,13] on page 4.

- Please check the formatting of each reference. Please check NLM abbreviations of journal names, for example: Reference 4, Journal of Dental Research should be J Dent Res and in Reference 13, PLOS Med should be PLoS Med.

PRESS

Sincerely, 

Caitlin Moyer, Ph.D. 

Associate Editor 

PLOS Medicine